# Challenges in antenatal care utilization in Kandahar, Afghanistan: A cross-sectional analytical study

Bilal Ahmad Rahimi[1,2]*, Enayatullah Mohamadi[3], Muhibullah Maku[3], Mohammad Dawood Hemat[3], Khushhal Farooqi[4], Bashir Ahmad Mahboobi[1], Ghulam Mohayuddin Mudaser[5], Walter R. Taylor[6]

1 Faculty of Medicine, Department of Pediatrics, Kandahar University, Kandahar, Afghanistan, 2 Head of Research Unit, Faculty of Medicine, Kandahar University, Kandahar, Afghanistan, 3 Faculty of Medicine, Department of Public Health, Kandahar University, Kandahar, Afghanistan, 4 Faculty of Medicine, Department of Dermatology, Kandahar University, Kandahar, Afghanistan, 5 Faculty of Medicine, Department of Histopathology, Kandahar University, Kandahar, Afghanistan, 6 Senior Clinical Research Fellow, Mahidol Oxford Tropical Medicine Clinical Research unit (MORU), Mahidol University, Bangkok, Thailand

* drbilal77@yahoo.com

## Abstract

### Background

Quality antenatal care (ANC) is one of the four pillars of safe motherhood initiatives and improves the survival and health of mother and neonate. The main objective of this study was to assess the barriers in the utilization of ANC services in Kandahar, Afghanistan.

### Methods

This was a cross-sectional analytical study conducted over one year from December 2018–November 2019. Data were analyzed by descriptive statistics, Chi squared, and binary logistic regression.

### Results

A total of 1524 women were recruited in this study with mean age of 30.3 years. Of these women, 848 (55.6%) were rural dwellers, 1450/1510 (96.0%) were illiterate, 438/608 (72.0%) belonged to low-income families, 1112/1508 (73.7%) lived in joint families, 1420/1484 (95.7%) lived in a house of >10 inhabitants, while 388/1494 (26.0%) had attended had at least one ANC visit during their last pregnancy. On univariate analysis, the main barriers in the utilization of ANC services were living in rural areas, being illiterate, having lower socio-economic status, remoteness of the health facility from home, bad behavior of clinic personnel, and unplanned pregnancy. Only lower socio-economic status and bad behavior of clinic personnel were independent explanatory variables in the regression model.

**Data Availability Statement:** All relevant data are within the paper and Supporting Information files.

**Funding:** This study did not receive any specific funding. WRT is partially-funded by Wellcome

under grant 220211. For the purpose of Open Access, the author has applied a CC BY public copyright licence to any Author Accepted Manuscript version arising from this submission.

**Competing interests:** The authors have declared that no competing interests exist.

## Conclusions

Utilization of ANC services is inadequate in Kandahar province. Improving clinic staff professional behavior and status of women by expanding educational opportunities, and enhancing community awareness of the value of ANC are recommended.

## Introduction

Globally, maternal mortality is one of the main public health issues [1]. It has been shown that better antenatal care (ANC) services improve the survival and health of both mothers and newborns [2]. World Health Organization (WHO) recommends at least eight ANC visits during pregnancy starting with the first visit at 12 weeks of gestational age (GA), then at 20, 26, 30, 34, 36, 38 and 40 weeks [3]. Afghanistan is among the top ten countries that contribute to more than half of the global maternal deaths [4,5]. In 2015 survey, only 59% of the pregnant women in Afghanistan attended at least one ANC visit [6].

In 2014, only 52% of the pregnant women attended four ANC visits in developing countries, where the mean MMR of 230/100,000 live births is 14 times greater than in developed countries [7]. Data show that when pregnant women in low- and middle-income countries (LMICs) receive better ANC from health facilities, most of the maternal and newborn deaths and pregnancy-related complications are prevented [8–11], e.g., neonatal deaths were 55% lower in women who had attended four ANC visits [12]. Moreover, ANC visits may detect early previously undiagnosed maternal morbidity as well as pregnancy related complications like eclampsia, small pelvis, and placenta previa [13,14]. ANC visits also provide a good opportunity for educating pregnant women about the warning symptoms and signs of common problems during pregnancy, healthy nutrition for the mother and newborn, and contraception for family planning [15].

Several factors affect ANC utilization in LMICs, including access to ANC, quality of ANC, socio-economic status, maternal education, demographic factors (e.g., maternal age and occupation), beliefs/knowledge about ANC, cultural beliefs, and previous obstetric history like unplanned pregnancy and parity [16–18].

Afghanistan is a low-income country and has been at war for several decades. As a result, the country faces significant challenges such as increasing poverty, continued political instability, and a devastated health infrastructure [5]. Afghanistan's challenges have been unique in consideration of ongoing conflicts for the last 45 years. These conflicts have severely affected not only the capacity of the health services to deliver quality ANC but also the broader disruption to the social determinants of health. Nevertheless, all public health facilities continue to offer free medical care, including ANC visits. According to the WHO, Afghanistan is one of the worst countries for pregnant women with a maternal mortality ratio (MMR) of 638 deaths/100,000 live births in 2017 [19]. Comparatively, in 2017, Somalia and Yemen which are also countries with devastating civil war, MMR was 829 and 164 deaths/100,000 live births, respectively [19]. The 2002 Reproductive Age Mortality Survey (RAMOS) conducted in Afghanistan estimated the MMR to be 1,600 deaths/100,000 live births [20]. Although The 2010 Afghanistan Mortality Survey (AMS) estimated the MMR to be 327/100,000 live births [21], the result of this survey was controversial and not acceptable [22]. Finally, the Afghanistan Demographic and Health Survey (AfDHS) 2015 reported that the pregnancy related mortality ratio (all maternal deaths during pregnancy, child birth, or within two months after pregnancy) was 1,291 maternal deaths per 100,000 live births.[6]. According to AfDHS data,

Afghanistan has the highest MMR in the world. Unfortunately, Afghanistan did not achieve the goal 5 of the Millennium Development Goals (MDGs) which was to reduce MMR to 75% by the year 2015 [23]. Also, the target 3.1 of the Sustainable Development Goals (SDGs) does not seem to be achieved which was to decreased MMR to <70/100,000 live births by the year 2030 [24]. In Afghanistan, ANC services are free at all public healthcare facilities. These facilities are provided by skilled healthcare staff including doctors, midwives, nurses, auxiliary midwives, and community health workers. Contrary, similar ANC services at private healthcare facilities are chargeable to the patients [6,25].

Studies from different parts of Afghanistan have revealed that the main factors affecting ANC utilization were level of maternal education, place of residence, previous health education on safe motherhood, media exposure, socio-economic status, availability of transport, and the behavior of healthcare personnel when seeing pregnant women [26–30]. There are very little published data regarding ANC utilization from Kandahar province which are solely limited to Kandahar city only [31,32]. We, therefore, investigated barriers in the ANC utilization in Kandahar city and Daman district located outside Kandahar city.

## Materials and methods

### Study design and period

This was a cross-sectional study questionnaire-based study that took place over 12 months from December 2018–November 2019.

### Study site and population

Kandahar province was selected for research due to the fact that it is one of the most unsecure provinces of Afghanistan. This study was conducted in four public health clinics in Kandahar city (Amir Jan comprehensive health center [CHC], Shams-ul-Haq Kakar CHC, Al-Khidmat CHC, and Nazo Ana CHC) and two public health clinics in Daman district (Mandisar CHC and Khoshab sub-health center). These health clinics were randomly selected using lottery-method. Daman district is a rural area adjoining Kandahar city. The sampling population consisted of all married women who attended any of the above-mentioned clinics for any reason (not only women attending ANC visit) and reported a pregnancy in the last one year.

### Primary objective

To assess the barriers in the utilization of ANC services in Kandahar Province, Afghanistan.
 **Inclusion criteria.**

- Married women who had given birth in the past one year prior to the study.

- Permanent residents for more than five years.

### Exclusion criteria

- Unmarried pregnant female. These females are excluded due to the facts that extramarital pregnancies are rare and also considered very big sin in the Afghan society. If the family members get information of extramarital pregnancy, there is a fear that the female can be tortured or even killed.

- Patients who refused to take part in the study.

## Sample size calculations

The sample size was based on the precision method and was calculated using Stata 15 (College Station, Texas, USA). Assuming an 85% response to a given question with a precision of 2%, the calculated sample size was 1440 females; in the event we analyzed 1524 females.

## Ethical considerations

Written informed consent was obtained from all the participants prior to the study. Ethical approval was taken from Kandahar University Ethics Committee with the approval number of 244/1397.

## Data collection and analysis

Data were collected from the respondents in a structured questionnaire developed based on relevant literature in a face-to-face interview. Initially, the questionnaire was drafted in English language. Later, it was translated into the local language (Pashto) by experts. Before the study, the questionnaire was pretested on 15 pregnant women attending ANC services in Shams-ul-Haq Kakar CHC with the aim of revising the poorly structured questions. The data were collected by trained female doctors and nurses using an exit interview with pregnant women. To ensure consistency, the data collection process was strictly supervised by principal investigator.

Data were analyzed with SPSS version 22 (Chicago, IL, USA) by descriptive statistics (proportions, means, and standard deviations). Chi squared (using crude odd ratio [COR]) was used to compare proportional data and 't' tests and their nonparametric equivalents were used to analyze continuous data. All variables that were statistically significant in univariate analyses were assessed for independence in a binary logistic regression (using adjusted odd ratio [AOR]) to determine the factors affecting the utilization of ANC services. A $P$-value of $<0.05$ was considered statistically significant.

Receiving antenatal care was defined as a pregnant woman having at least one antenatal care check-up during their last pregnancy from health facility [33].

Distance to the nearby health facility was defined according to the history the mothers gave: near if mothers accessed the clinic $<$ 30 minutes while "remote or far away" was defined as $\geq$30 minutes [34].

## Study variables and their indicators

- Socio-demographic characteristics included age, socio-economic status, employment, literacy level, residence, parity (number of babies delivered), and number of family members living in the same house.

- Attitudes and practice included clinic staff behavior, at least one ANC visit done during last pregnancy, number of ANC visits during last pregnancy, and reason of not attending ANC visit.

These above-mentioned variables have been reported to be the barriers in the utilization of ANC services in Afghanistan [31] and other parts of the world [16–18].

## Definitions

**Socio-economic status** [31,35]

 Low income = $<$ 2500 Afghanis ($<$ 30 USD) per month.
 Middle income = 2500–20,000 Afghanis (30–250 USD) per month.

High income = > 20,000 Afghanis (> 250 USD) per month.

### Negative clinic staff behavior

Presence of one or more of the following behaviors: hostility, aggressiveness, rudeness, disrespect, physical abuse or bullying toward the patients [25,26,36,37].

## Results

Of the 1610 pregnant women who had visited their local ANC within 1 year, 1524 married women agreed to participate in the study. Their mean age was 30.3 years (range 16 to 50). More than half of them, 62.3% (950/1524) were aged between 21–30 years and 848 (55.6%) were rural dwellers.

Almost all, 1520/1524 (99.7%), were housewives, 1450/1510 (96.0%) were illiterate and 438/608 (72.0%) came from low-income families. The majority, 1112/1508 (73.7%), lived with extended families and 1420/1484 (95.7%) lived in households of >10 inhabitants (Table 1). Only 388/1494 (26.0%) attended the ANC at least once and main reason (511/1106 [46.2%]) for poor attendance was remoteness of the health facility from their home (Table 2).

In the univariate analysis, significant barriers in the utilization of ANC services were living in rural areas (COR 1.4), being illiterate (COR 2.4), low socio-economic status (COR 1.5),

**Table 1. Socio-demographic characteristics of the study participants.**

| Variable | Number (n = 1524) | Percentage (%) |
|---|---|---|
| Age (years) | | |
| ≤20 | 92 | 6.0 |
| 21–30s | 950 | 62.3 |
| 31–40 | 406 | 26.7 |
| >40 | 76 | 5.0 |
| Socio-economic status (n = 1516) | | |
| Low income | 528 | 34.8 |
| Middle income | 782 | 51.6 |
| High income | 206 | 13.6 |
| Employment | | |
| Employed | 4 | 0.3 |
| Housewife | 1520 | 99.7 |
| Literacy level (n = 1510) | | |
| Literate | 60 | 4.0 |
| Illiterate | 1450 | 96.0 |
| Number of babies delivered (n = 1450) | | |
| 1 | 152 | 10.5 |
| 2–5 | 876 | 60.4 |
| >5 | 422 | 29.1 |
| Number of children (n = 1400) | | |
| 1 | 90 | 6.4 |
| 2–5 | 810 | 57.9 |
| >5 | 500 | 35.7 |
| Age of last child (n = 1352) | | |
| ≤1 year | 286 | 21.2 |
| >1 year | 1066 | 78.8 |
| Type of family (n = 1508) | | |
| Nuclear | 396 | 26.3 |
| Joint | 1112 | 73.7 |
| Number of family members living in the same house (n = 1484) | | |
| <5 | 18 | 1.2 |
| 5–10 | 46 | 3.1 |
| >10 | 1420 | 95.7 |

**Table 2. ANC-related and other variables in study participants.**

| Variable | Number (*n*) | Percentage (%) |
|---|---|---|
| At least one ANC visit done during last pregnancy (n = 1494) | | |
| Yes | 388 | 26.0 |
| No | 1106 | 74.0 |
| Number of ANC visits during last pregnancy (n = 388) | | |
| Once | 35 | 9.0 |
| 2–4 times | 225 | 58.1 |
| >4 times | 128 | 32.9 |
| Reason for not attending ANC visit (n = 1106) | | |
| Clinic is far away | 511 | 46.2 |
| No medicine in clinic | 362 | 32.7 |
| No night duty staffs in clinic | 57 | 5.2 |
| Clinic staff do not have good behavior | 22 | 2.0 |
| Family does not allow | 154 | 13.9 |
| Clinic present near home (n = 1510) | | |
| Yes | 1100 | 72.8 |
| No | 410 | 27.2 |
| Distance from house to clinic (walking) (n = 1488) | | |
| <30 minutes | 448 | 30.1 |
| 30–60 minutes | 660 | 44.4 |
| 61 minutes–2 hours | 348 | 23.4 |
| >2 hours | 32 | 2.2 |
| Clinic staff behavior (n = 1476) | | |
| Good | 1114 | 75.5 |
| Negative (not good) | 362 | 24.5 |
| Planned pregnancy (n = 1000) | | |
| Yes | 804 | 80.4 |
| No | 196 | 19.6 |
| Method used to make drinking water safe (n = 1472) | | |
| Boil | 1106 | 75.1 |
| Add bleach/chlorine | 160 | 10.9 |
| Strain through a cloth | 128 | 8.7 |
| Use water filter | 78 | 5.3 |

ANC, Ante-natal care; n, number.

remoteness of health facility from home (COR 1.8), bad behavior of clinic personnel (COR 3.2), and an unplanned pregnancy (COR 1.5). By logistic regression only two statistically significant barriers to ANC utilization remained: bad behavior of clinic personnel (AOR 9.4) and low socio-economic status (AOR 2.3). A higher literacy level was associated with greater utilization of ANC (Table 3).

## Discussion

In this large survey from Kandahar, we collected data from 1524 women and identified two key independent factors for poor ANC utilization: poor professional behavior by clinic staff and low socioeconomic status. By contrast, a higher level of literacy was associated with greater utilization.

Very few ANC studies have been conducted in Afghanistan. Most published articles are based on the retrospective survey data conducted by Afghanistan MoPH; they show several overlapping reasons for poor ANC utilization [5,25–27,30,38]. The main independent factors associated with no ANC visits were young maternal age (15–19 years), being a working mother, and the decision for healthcare being taken by the husband [25]. In Kabul and Ghazni (a province SW of Kabul), underuse of ANC services was associated with low maternal

**Table 3. Univariate analyses and logistic regression of barriers to the utilization of ante natal care services.**

| Variable | Total, n (%) | ANC visit(s) done | | COR (95% CI) | P-value | AOR (95% CI) | P-value |
|---|---|---|---|---|---|---|---|
| | | Yes, n (%) | No, n (%) | | | | |
| Age (years) (n = 1494) | | | | | 0.189 | | |
| >30 | 476 (31.9) | 134 (28.2) | 342 (71.8) | 1 | | | |
| ≤ 30 | 1018 (68.1) | 254 (25.0) | 764 (75.0) | 0.8 (0.7–1.1) | | | |
| Place of living (n = 1494) | | | | | 0.003 | | 0.105 |
| Urban | 666 (44.6) | 198 (29.7) | 468 (70.3) | 1 | | 1 | |
| Rural | 828 (55.4) | 190 (22.9) | 638 (77.1) | 1.4 (1.1–1.8) | | 2.3 (0.8–6.4) | |
| Literacy level (n = 1480) | | | | | 0.022 | | 0.002 |
| Literate | 60 (4.0) | 27 (45.0) | 33 (55.0) | 1 | | 1 | |
| Illiterate | 1420 (96.0) | 359 (25.3) | 1061 (74.7) | 2.4 (1.1–5.0) | | 0.1 (0.0–0.4) | |
| Number of children (n = 1380) | | | | | 0.076 | | |
| >5 | 492 (35.7) | 120 (24.4) | 372 (75.6) | 1 | | | |
| ≤5 | 888 (64.3) | 256 (28.8) | 632 (71.2) | 1.3 (1.0–1.6) | | | |
| Age of last child (n = 1344) | | | | | <0.001 | | |
| ≤1 year | 284 (21.1) | 48 (16.9) | 236 (83.1) | 1 | | | |
| >1 year | 1060 (78.9) | 310 (29.2) | 750 (70.8) | 0.5 (0.4–0.7) | | | |
| Type of family (n = 1478) | | | | | 0.006 | | |
| Nuclear | 384 (26.0) | 80 (20.8) | 304 (79.2) | 1 | | | |
| Joint | 1094 (74.0) | 306 (28.0) | 788 (72.0) | 0.7 (0.5–0.9) | | | |
| Number of family members living in the same house (n = 116) | | | | | 0.120 | | |
| <5 | 18 (15.5) | 2 (11.1) | 16 (88.9) | 1 | | | |
| ≥ 5 | 98 (84.5) | 28 (28.6) | 70 (71.4) | 0.3 (0.1–1.4) | | | |
| Socio-economic status (n = 1488) | | | | | 0.001 | | 0.044 |
| Low income | 514 (34.5) | 114 (22.2) | 400 (77.8) | 1 | | 1 | |
| Middle/High income | 974 (65.5) | 272 (27.9) | 702 (72.1) | 1.5 (1.2–1.9) | | 2.3 (1.0–5.4) | |
| Clinic present near home (n = 1480) | | | | | <0.001 | | 0.213 |
| No | 404 (27.3) | 74 (18.3) | 330 (81.7) | 1 | | 1 | |
| Yes | 1076 (72.7) | 314 (29.2) | 762 (70.8) | 1.8 (1.4–2.4) | | 0.5 (0.2–1.4) | |
| Distance from house to clinic (n = 1460) | | | | | 0.966 | | |
| <30 minutes | 440 (30.1) | 116 (26.4) | 324 (73.6) | 1 | | | |
| ≥30 minutes | 1020 (69.9) | 270 (26.5) | 750 (73.5) | 1.0 (0.8–1.3) | | | |
| Clinic staff behavior (n = 1458) | | | | | <0.001 | | 0.001 |
| Not good | 354 (24.3) | 44 (12.4) | 310 (87.6) | 1 | | 1 | |
| Good | 1104 (75.7) | 342 (31.0) | 762 (69.0) | 3.2 (2.2–4.4) | | 9.4 (2.4–36.7) | |
| Planned pregnancy (n = 986) | | | | | 0.039 | | 0.274 |
| No | 194 (19.7) | 34 (17.5) | 160 (82.5) | 1 | | 1 | |
| Yes | 792 (80.3) | 194 (24.5) | 598 (75.5) | 1.5 (1.0–2.3) | | 1.8 (0.6–5.2) | |

ANC, Ante-natal Care; AOR, Adjusted Odds Ratio; CI, Confidence Interval; COR, Crude Odds Ratio; n, number.

motivation, family decision, notably the mother-in-law and husband not consenting to the ANC visit, lower socio-economic status, and transportation challenges [26]. A recent community-based cross-sectional study in Kandahar city revealed that main determinants of underuse of ANC utilization were illiteracy, unplanned pregnancy, and living in poorer districts of the city [31].

The main factors determining ANC utilization from different developing countries include illiteracy, lower socio-economic status, remoteness of the health facility, transportation challenges, and living in rural areas [39–45]. A study based on the evidence from demographic health surveys in sub-Saharan Africa revealed that main barriers to the utilization of ANC services were decreased literacy level, living in rural areas, low socio-economic status, and not getting permission to visit the health facility [46]. A systematic review and meta-analysis of 15 observational studies in Ethiopia concluded that rural residence, illiteracy of woman or her husband, and unplanned pregnancy were the main barriers in ANC services utilization [2].

We showed that poor professional behavior of clinic staff was a more important factor than socioeconomic status leading to a reluctance of women to attend ANCs. Another study in Afghanistan also reported significant dissatisfaction with the behavior of health personnel, which included verbal and physical abuse [26]. Health care dissatisfaction is also reported from Ethiopia. In Jimma (central Ethiopia), 67.1%, 49.9%, and 37.8% of the pregnant women were dissatisfied with the physical environment of the ANC, quality of care, and organization of health care [36]. In Kuala Lumpur, Malaysia, 81.3%, 61.7%, and 51.3% of the ANC attendants were not satisfied with the continuity of care, accessibility, and convenience in the antenatal clinic [37] and most women (60.7%) attending ANCs specializing in the prevention of maternal to mother transmission of HIV in Benin city, Nigeria, were dissatisfied with the counselling service [47].

Although ANC is free of charge in all public health facilities in Afghanistan, women may seek care in the private sector to avoid the disadvantages of the public health system such as unprofessional staff behavior, poor infrastructure, absent ANC staff, and a shortage of medications, especially in rural areas.

In the univariate analysis, rural women were less likely to attend ANC visits than their urban counterparts but this was not significant in the regression model; this result may have been due to reduced power in the model. By contrast, several studies report rural dwellers are less likely to attend ANCs in India [48], Nepal [49], Indonesia [50,51], Ghana [52], Sudan [53], and Ethiopia [54]. We found that a higher level of maternal literacy was associated independently with good ANC attendance, similar to many other studies [16,39,51,55–57]. A higher level of maternal education may mean that such women should be better informed of the benefits of ANC and better able to decide for themselves. If they are urban dwellers, they will also have greater access to health information and have greater accessibility to clinics [58,59]. Although remoteness from the nearest ANC was not a significant factor in our regression model, it was a key factor in a range of studies from Kenya [45], Rwanda [60], Ethiopia [61], Haiti [62], and Indonesia [63]; however, one Indian study found it was not a significant factor [39].

## Limitations of the study

Although large, our study had limitations. We interviewed women once and so did not take into account risk factors that may change over time in attending the ANC. Another limitation could be recall bias because questions to women were about events in the past. Moreover, we did not ascertain the clinical course and outcome of the pregnancy in question or the presence of comorbidities.

## Conclusions

The utilization of ANC services is very poor in Kandahar province. Although several intuitive reasons were identified in the univariate analysis, only poor staff behavior and low socioeconomic status were independent factors. More research is needed to explore other factors affecting women's decision to forego ANC attendance like a previously complicated pregnancy, the presence of comorbidities, and experience with the private sector. Ways to improve the professionalism of clinic staff is needed urgently. Assessing simple, low-tech interventions like health messaging and the acceptability of training 'bush' midwives (also known as traditional birth attendants) to carry out simple pregnancy assessments in the field should be conducted. Moreover, improving the status of women by expanding educational opportunities, strengthening promotion of antenatal and delivery care by enhancing community awareness of the importance of antenatal, natal, and post-natal care are recommended.

## Supporting information

**S1 File. Questionnaire used for the research.**
(DOCX)

**S2 File. SPSS file with a part of the research data.**
(SAV)

## Acknowledgments

We present our highest and sincere thanks to the authorities Faculty of Medicine, Kandahar University, Directorate of Public Health, and staff members of health facilities. We are also very thankful of all the women who consented to take part in our study.

## Author Contributions

**Conceptualization:** Bilal Ahmad Rahimi, Enayatullah Mohamadi, Muhibullah Maku, Mohammad Dawood Hemat, Khushhal Farooqi, Bashir Ahmad Mahboobi, Walter R. Taylor.

**Data curation:** Bilal Ahmad Rahimi, Ghulam Mohayuddin Mudaser.

**Formal analysis:** Bilal Ahmad Rahimi, Walter R. Taylor.

**Investigation:** Bilal Ahmad Rahimi, Enayatullah Mohamadi, Khushhal Farooqi, Ghulam Mohayuddin Mudaser.

**Methodology:** Bilal Ahmad Rahimi, Bashir Ahmad Mahboobi, Ghulam Mohayuddin Mudaser.

**Project administration:** Bilal Ahmad Rahimi, Mohammad Dawood Hemat.

**Software:** Bashir Ahmad Mahboobi.

**Supervision:** Bilal Ahmad Rahimi, Muhibullah Maku, Ghulam Mohayuddin Mudaser.

**Writing – original draft:** Bilal Ahmad Rahimi.

**Writing – review & editing:** Bilal Ahmad Rahimi, Enayatullah Mohamadi, Muhibullah Maku, Mohammad Dawood Hemat, Khushhal Farooqi, Bashir Ahmad Mahboobi, Ghulam Mohayuddin Mudaser, Walter R. Taylor.

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
