## [Decision Letter · Decision Letter 0]

26 Apr 2022

PONE-D-21-18840Title of the Article: Challenges in antenatal care utilization in Kandahar, Afghanistan: A cross-sectional analytical study.PLOS ONE

Dear Dr. Rahimi,

Thank you for submitting your manuscript to PLOS ONE. I sincerely apologise for the unusually delayed review timeframe. After careful consideration, we feel that it has merit but does not fully meet PLOS ONE’s publication criteria as it currently stands. Therefore, we invite you to submit a revised version of the manuscript that addresses the points raised during the review process. (Please see attached report from the reviewer.) In addition to the concerns that the reviewer has raised, please address the following editorial concerns:- In the Methods section, please provide detailed information about how the questionnaire was developed, and how or whether it was pre-tested and validated.- Please provide the method by which the sample size was calculated.

Please note that we have only been able to secure a single reviewer to assess your manuscript. We are issuing a decision on your manuscript at this point to prevent further delays in the evaluation of your manuscript. Please be aware that the editor who handles your revised manuscript might find it necessary to invite additional reviewers to assess this work once the revised manuscript is submitted. However, we will aim to proceed on the basis of this single review if possible.

We look forward to receiving your revised manuscript.

Kind regards,

Emily Chenette

Editor in Chief

PLOS ONE

Journal Requirements:

2. Please include additional information regarding the survey or questionnaire used in the study and ensure that you have provided sufficient details that others could replicate the analyses. For instance, if you developed the survey or questionnaire as part of this study and it is not under a copyright more restrictive than CC-BY, please include a copy, in both the original language and English, as Supporting Information. If the questionnaire is published, please provide a citation to the (1) questionnaire and/or (2) original publication associated with the questionnaire.

3. Please revise the title of your manuscript to remove "Title of the Article:" in the submission system.

4. Thank you for stating the following financial disclosure: "There was no financial support for this research from any source."

5. Thank you for stating the following in your Competing Interests section:  "There are no competing interests."

Reviewers' comments:

Reviewer's Responses to Questions

**Comments to the Author**

1. Is the manuscript technically sound, and do the data support the conclusions?

Reviewer #1: Partly

2. Has the statistical analysis been performed appropriately and rigorously? 

Reviewer #1: Yes

3. Have the authors made all data underlying the findings in their manuscript fully available?

Reviewer #1: No

4. Is the manuscript presented in an intelligible fashion and written in standard English?

Reviewer #1: Yes

5. Review Comments to the Author

Reviewer #1: I was pleased to read this important paper from a high maternal burden country. The authors are provided enclosed comments for a chance to improve the paper. The paper has potential to be published but key areas where the authors may like to focus are:formatting, background context and presentation of information.

6. PLOS authors have the option to publish the peer review history of their article (what does this mean?). If published, this will include your full peer review and any attached files.

Reviewer #1: **Yes: **Dr Danish Ahmad

---

## [Author Response · Author response to Decision Letter 0]

3 Jun 2022

PONE-D-21-18840R1

Title of the Article: Challenges in antenatal care utilization in Kandahar, Afghanistan: A cross-sectional analytical study.

Bilal Ahmad Rahimi

Dear Dr. Rahimi,

We've checked your submission and before we can proceed, we need you to address the following issues:

1. Please upload a Response to Reviewers letter which should include a point by point response to each of the points made by the Editor and / or Reviewers. (This should be uploaded as a 'Response to Reviewers' file type.) Please follow this link for more information: http://blogs.PLOS.org/everyone/2011/05/10/how-to-submit-your-revised-manuscript/

Answer: A "Response to reviewers" file has been uploaded.

2. Please include additional information regarding the survey or questionnaire used in the study and ensure that you have provided sufficient details that others could replicate the analyses. For instance, if you developed the survey or questionnaire as part of this study and it is not under a copyright more restrictive than CC-BY, please include a copy, in both the original language and English, as Supporting Information. If the questionnaire is published, please provide a citation to the (1) questionnaire and/or (2) original publication associated with the questionnaire.

Answer: OK. Now questionnaire has been uploaded as Supporting Information.

3. Please revise the title of your manuscript to remove "Title of the Article:" in the submission system.

Answer: Now "Title of the Article:" is removed in the submission system.

4. Thank you for stating the following financial disclosure: "There was no financial support for this research from any source."

Answer: We did not have any funding sources for our study. The publication fee for our article will be provided by MORU (Mahidol-Oxford Tropical Medicine Research Unit), Bangkok. MORU will pay only if the journal put the following statement in the funding part of the article:

“This study did not receive any specific funding. WR Taylor is part funded by Wellcome under grant 220211. For the purpose of Open Access, the author has applied a CC BY public copyright licence to any Author Accepted Manuscript version arising from this submission.”

I have mentioned this in the Cover letter too.

5. Thank you for stating the following in your Competing Interests section: "There are no competing interests."

Answer: We have added the following sentence in the Cover letter:

“The authors have declared that no competing interests exist.”.

Answer: As the data file is the property of Kandahar University Research Center, we are not allowed to share the data file with anyone. For this we contacted the research center. They provided a part of the data from the main SPSS data file. Now have uploaded it as the Supporting Information file.

Also, we mentioned this information in the Cover letter.

Thank you for submitting your work to PLOS ONE and supporting our mission of Open Science.

Kind regards,

Richard Ibañez Dilla

PLOS ONE

Many thanks

Bilal

---

## [Decision Letter · Decision Letter 1]

10 Jul 2022

PONE-D-21-18840R1

Challenges in antenatal care utilization in Kandahar, Afghanistan: A cross-sectional analytical study.

PLOS ONE

Dear Dr. Rahimi,

Thank you for submitting your manuscript to PLOS ONE. After careful consideration, we feel that it has merit but does not fully meet PLOS ONE’s publication criteria as it currently stands. Therefore, we invite you to submit a revised version of the manuscript that addresses the points raised during the review process.

The reviewer from the first round has reassessed the manuscript. Whilst they are overall happy with the amendments made, they have provided some additional suggestions, which can be found below.

Please also amend your Methods section to include details of how the questionnaire was developed, tested and validated.

We look forward to receiving your revised manuscript.

Kind regards,

Hanna Landenmark

Staff Editor

PLOS ONE

Journal Requirements:

Reviewers' comments:

Reviewer's Responses to Questions

**Comments to the Author**

1. If the authors have adequately addressed your comments raised in a previous round of review and you feel that this manuscript is now acceptable for publication, you may indicate that here to bypass the “Comments to the Author” section, enter your conflict of interest statement in the “Confidential to Editor” section, and submit your "Accept" recommendation.

Reviewer #1: (No Response)

2. Is the manuscript technically sound, and do the data support the conclusions?

Reviewer #1: Yes

3. Has the statistical analysis been performed appropriately and rigorously? 

Reviewer #1: Yes

4. Have the authors made all data underlying the findings in their manuscript fully available?

Reviewer #1: Yes

5. Is the manuscript presented in an intelligible fashion and written in standard English?

Reviewer #1: Yes

6. Review Comments to the Author

Reviewer #1: Dear Authors,

Thank you for submitting an improved version

The response to my previous comments has been attempted but has not been satisfactorily completed. The background section for example starts with a summary of MDG's and the SDG goal' which is not needed and does not add value to the paper. Rather,Afghanistan's MMR progress in the SDG and MDG is important to highlight. Similarily, the study variables on page 8 lines 193 need better explanation of why they were chosen with links to the litertaure. The definitions sections in page 9 also needs work-Please provide narrative explanation of these definitions along with links to the litertaure. Formatting of table 3 remains an issue .I would encourage the authors to make these changes in depth as the paper provides important findings

7. PLOS authors have the option to publish the peer review history of their article (what does this mean?). If published, this will include your full peer review and any attached files.

Reviewer #1: **Yes: **Dr Danish Ahmad

---

## [Author Response · Author response to Decision Letter 1]

3 Aug 2022

PONE-D-21-18840R1

Title of the Article: Challenges in antenatal care utilization in Kandahar, Afghanistan: A cross-sectional analytical study.

Bilal Ahmad Rahimi

• Please also amend your Methods section to include details of how the questionnaire was developed, tested and validated.

Answer: Now details have been added in the “Materials and methods” section (lines 181-189) to clarify how the questionnaire was developed, tested and validated.

Review Comments to the Author

Reviewer #1: Dear Authors,

Thank you for submitting an improved version.

The response to my previous comments has been attempted but has not been satisfactorily completed. 

1. The background section for example starts with a summary of MDG's and the SDG goal' which is not needed and does not add value to the paper. Rather, Afghanistan's MMR progress in the SDG and MDG is important to highlight. 

Answer: Summary of MDGs and SDGs goals have been removed from the “Introduction” section. Afghanistan's MMR progress in the SDG and MDG is now added in the second last paragraph of the “Introduction”.

2. Similarly, the study variables on page 8 lines 193 need better explanation of why they were chosen with links to the literature. 

Answer: Now the study variables on page 8 have explained why they were chosen and also liks to the literatures have been provided.

3. The definitions sections in page 9 also needs work-Please provide narrative explanation of these definitions along with links to the literature. 

Answer: Now the definitions section has been explained narratively and links are provided.

4. Formatting of table 3 remains an issue. I would encourage the authors to make these changes in depth as the paper provides important findings.

Answer: I formatted table 3. To make table 3 clearer and easy to understand, I merged tables 3 and 4 as one table. Now this table contain both Chi-square analyses and logistic regression, making it very easy to understand.

Many thanks

Bilal

---

## [Decision Letter · Decision Letter 2]

26 Sep 2022

PONE-D-21-18840R2Challenges in antenatal care utilization in Kandahar, Afghanistan: A cross-sectional analytical study.PLOS ONE

Dear Dr. Rahimi,

Thank you for submitting your manuscript to PLOS ONE. After careful consideration, we feel that it has merit but does not fully meet PLOS ONE’s publication criteria as it currently stands. Therefore, we invite you to submit a revised version of the manuscript that addresses the points raised during the review process.

While the paper provides important insights into challenges of ANC service utilization in Kandhar, Afghanistan; please address key issues mentioned below in your revised submission.

We look forward to receiving your revised manuscript.

Kind regards,

Monalisha Sahu

Academic Editor

PLOS ONE

Journal Requirements:

Additional Editor Comments (if provided):

Dear Authors,

Thanks for submitting your research with PLOS One. While the paper provides important insights to challenges of ANC service utilization in Kandhar, Afghanistan; there are certain key issues which must be addressed beforehand for possible publication.

Introductions:

• The first paragraph could be shortened and should be written more focussed on Afghanistan; the irrelevant information can be removed.

• 92 Generalized sentences like- ‘International community has always tried to reduce maternal mortality’ should be converted to more specific ones.

Methodology:

• 152.Study Design should be mentioned more specifically (almost all quantitative studies are questionnaire based)

• 156. Why these four clinics were chosen in Kandahar city should be explained properly.

• 160. Why Kandhar province was selected for the studyshould move up in the study methodology, probably to line No. 156.

• 170.Why unmarried female were excluded and how ethical consideration were met for them may be discussed briefly?

• 203. The Independent variable section n methodology does not mention Breastfeeding time of last child or any other variable related to ‘Using methods to make drinking water safe’; but they have been included in the result tables 2 &3. Also, how these variables can affect ANC services utilization can be briefly addressed for wide comprehension. If deemed not suitable these variables should be removed from the result tables.

Formatting of intext citation should be checked and corrected as per journal policy.

Reviewers' comments:

Reviewer's Responses to Questions

**Comments to the Author**

1. If the authors have adequately addressed your comments raised in a previous round of review and you feel that this manuscript is now acceptable for publication, you may indicate that here to bypass the “Comments to the Author” section, enter your conflict of interest statement in the “Confidential to Editor” section, and submit your "Accept" recommendation.

Reviewer #1: All comments have been addressed

Reviewer #2: (No Response)

2. Is the manuscript technically sound, and do the data support the conclusions?

Reviewer #1: Yes

Reviewer #2: Yes

3. Has the statistical analysis been performed appropriately and rigorously? 

Reviewer #1: Yes

Reviewer #2: Yes

4. Have the authors made all data underlying the findings in their manuscript fully available?

Reviewer #1: (No Response)

Reviewer #2: Yes

5. Is the manuscript presented in an intelligible fashion and written in standard English?

Reviewer #1: Yes

Reviewer #2: Yes

6. Review Comments to the Author

Reviewer #1: Dear Authors,

Thank you for a much improved version. The paper reads better but still has formatting issues linked to intext references.For sequential references used intext such as1,2,3,4 please use [1-4]. I advice the authors to spend time reviewing the paper and checking for foramtting issues. As this is the only revision, the paper stands in a good place to be published if addressed and deemed by the editor

Reviewer #2: (No Response)

7. PLOS authors have the option to publish the peer review history of their article (what does this mean?). If published, this will include your full peer review and any attached files.

Reviewer #1: **Yes: **Dr Danish Ahmad

Reviewer #2: No

---

## [Author Response · Author response to Decision Letter 2]

26 Sep 2022

PONE-D-21-18840R1

Title of the Article: Challenges in antenatal care utilization in Kandahar, Afghanistan: A cross-sectional analytical study.

Bilal Ahmad Rahimi

Review Comments to the Author

Dear Authors,

Thanks for submitting your research with PLOS One. While the paper provides important insights to challenges of ANC service utilization in Kandahar, Afghanistan; there are certain key issues which must be addressed beforehand for possible publication.

Introductions:

• The first paragraph could be shortened and should be written more focussed on Afghanistan; the irrelevant information can be removed.

• Answer: OK. Now the first paragraph has been shortened, irrelevant information has been removed, and now focussed on Afghanistan.

• 92 Generalized sentences like- ‘International community has always tried to reduce maternal mortality’ should be converted to more specific ones.

• Answer: OK. Now these generalized sentences have been removed and converted into more specific sentences.

Methodology:

• 152.Study Design should be mentioned more specifically (almost all quantitative studies are questionnaire based)

• Answer: OK. Thanks for the comment. To make it clearer and specific, now I changed it to “Cross-sectional questionnaire-based study” .

• 156. Why these four clinics were chosen in Kandahar city should be explained properly.

• Answer: OK. Now it is clearly explained. “These health clinics were randomly selected using lottery-method.”

• 160. Why Kandhar province was selected for the study should move up in the study methodology, probably to line No. 156.

• Answer: OK. As per reviewer comment, now this sentence is moved up to line 156.

• 170.Why unmarried female were excluded and how ethical consideration were met for them may be discussed briefly?

• Answer: Now the following sentences have been added in exclusion criteria after “unmarried females”:

“These females are excluded due to the facts that extramarital pregnancies are rare and also considered very big sin in the Afghan society. If the family members get information of extramarital pregnancy, there is a fear that the female can be tortured or even killed.”

• 203. The Independent variable section n methodology does not mention Breastfeeding time of last child or any other variable related to ‘Using methods to make drinking water safe’; but they have been included in the result tables 2 &3. Also, how these variables can affect ANC services utilization can be briefly addressed for wide comprehension. If deemed not suitable these variables should be removed from the result tables.

• Answer: OK. Thanks for the good point. As these 2 variables (i.e., “Breastfeeding time of last child” and “Using methods to make drinking water safe” do not affect ANC services utilization, now they have been removed from the tables.

Formatting of intext citation should be checked and corrected as per journal policy.

• Answer: OK. Thanks. Now the intext citation is thoroughly checked and corrected as per journal policy. For example, intext citation of 1,2,3,4 has been changed to 1-4.

Many thanks

Bilal

---

## [Editor Report · Decision Letter 3]

20 Oct 2022

Challenges in antenatal care utilization in Kandahar, Afghanistan: A cross-sectional analytical study.

PONE-D-21-18840R3

Dear Dr. Rahimi,

We’re pleased to inform you that your manuscript has been judged scientifically suitable for publication and will be formally accepted for publication once it meets all outstanding technical requirements.

Kind regards,

Monalisha Sahu

Academic Editor

PLOS ONE

Additional Editor Comments (optional):

The in text citation still needs correction
---

## [Editor Report · Acceptance letter]

11 Nov 2022

PONE-D-21-18840R3 

Challenges in antenatal care utilization in Kandahar, Afghanistan: A cross-sectional analytical study. 

Dear Dr. Rahimi:

I'm pleased to inform you that your manuscript has been deemed suitable for publication in PLOS ONE. Congratulations! Your manuscript is now with our production department. 

Kind regards, 

on behalf of

Dr. Monalisha Sahu 

Academic Editor

PLOS ONE